# Management of Recurrent Well-Differentiated Thyroid Carcinoma in the Neck: A Comprehensive Review [note 1]

**DOI:** 10.3390/cancers15030923

**Published:** 2023-02-01

**Authors:** Beatriz G. Cavalheiro, Jatin P. Shah, Gregory W. Randolph, Jesus E. Medina, Ralph P. Tufano, Mark Zafereo, Dana M. Hartl, Iain J. Nixon, Orlando Guntinas-Lichius, Vincent Vander Poorten, Fernando López, Avi Hefetz Khafif, Randall P. Owen, Ashok Shaha, Juan P. Rodrigo, Alessandra Rinaldo, Antti A. Mäkitie, Carl E. Silver, Alvaro Sanabria, Luiz P. Kowalski, Alfio Ferlito

**Affiliations:** 1Cancer Institute of São Paulo State, Department of Head and Neck Surgery, University of São Paulo Medical School, Sao Paulo 01246-903, Brazil; 2Head and Neck Service, Memorial Sloan Kettering Cancer Center, New York, NY 10065, USA; 3Department of Otolaryngology Head and Neck Surgery, Harvard Medical School, Boston, MA 02115, USA; 4Department of Otolaryngology and Head and Neck Surgery, University of Oklahoma College of Medicine, Oklahoma City, OK 73104, USA; 5Multidisciplinary Thyroid and Parathyroid Center, Head and Neck Endocrine Surgery, Sarasota Memorial Health Care System, Sarasota, FL 34239, USA; 6Department of Head & Neck Surgery, MD Anderson Cancer Center, Houston, TX 77030, USA; 7Department of Surgery, Gustave Roussy, 94800 Villejuif, France; 8Department of Otolaryngology Head and Neck Surgery, NHS Lothian, University of Edinburgh, Edinburgh EH8 9YL, UK; 9Department of Otorhinolaryngology-Head and Neck Surgery, Jena University Hospital, 07747 Jena, Germany; 10Otorhinolaryngology, Head and Neck Surgery, University Hospitals Leuven, 3000 Leuven, Belgium; 11ENT and Head and Neck Department, Hospital Universitario Central de Asturias, University of Oviedo, ISPA, IUOPA, 33011 Oviedo, Spain; 12Centro de Investigación Biomédica en Red de Cancer (CIBERONC), 28029 Madrid, Spain; 13Assuta Medical Center, Ben-Gurion University of the Negev, Tel Aviv 8436322, Israel; 14Section of Endocrine Surgery, Division of Surgical Oncology, Department of Surgery, Mount Sinai Hospital, Icahn School of Medicine, New York, NY 10029, USA; 15Head and Neck Service, Department of Surgery, Memorial Sloan-Kettering Cancer Center, New York, NY 10065, USA; 16School of Medicine, University of Udine, 33100 Udine, Italy; 17Department of Otorhinolaryngology-Head and Neck Surgery, Helsinki University Hospital, University of Helsinki, 00290 Helsinki, Finland; 18Department of Surgery, University of Arizona College of Medicine, Phoenix, AZ 85724, USA; 19Department of Surgery, School of Medicine, University of Antioquia, Medellín 0500100, Colombia; 20Department of Head and Neck Surgery and Otorhinolaryngology, AC Camargo Cancer Center, Sao Paulo 01509-001, Brazil; 21Department of Head and Neck Surgery, University of Sao Paulo Medical School, Sao Paulo 01246-903, Brazil; 22Coordinator of the International Head and Neck Scientific Group, 35125 Padua, Italy

**Keywords:** thyroid neoplasm, papillary thyroid cancer, metastasis, recurrence, surgery, neck dissection, radiotherapy, radiofrequency ablation, molecular targeted therapy, neoadjuvant therapy

## Abstract

**Simple Summary:**

Surgery is generally the treatment of choice for locoregional recurrences of well-differentiated thyroid carcinomas, but other therapies can be considered on an individual basis. These patients are expected to have prolonged survival, even with the possibility of long periods of active disease and the need for subsequent treatments. The present review intends to provide considerations regarding these therapeutic possibilities.

**Abstract:**

Surgery has been historically the preferred primary treatment for patients with well-differentiated thyroid carcinoma and for selected locoregional recurrences. Adjuvant therapy with radioactive iodine is typically recommended for patients with an intermediate to high risk of recurrence. Despite these treatments, locally advanced disease and locoregional relapses are not infrequent. These patients have a prolonged overall survival that may result in long periods of active disease and the possibility of requiring subsequent treatments. Recently, many new options have emerged as salvage therapies. This review offers a comprehensive discussion and considerations regarding surgery, active surveillance, radioactive iodine therapy, ultrasonography-guided percutaneous ablation, external beam radiotherapy, and systemic therapy for well-differentiated thyroid cancer based on relevant publications and current reference guidelines. We feel that the surgical member of the thyroid cancer management team is empowered by being aware and facile with all management options.

## 1. Introduction

Surgery is the primary modality of therapy for patients with well-differentiated thyroid carcinoma (WDTC) [1,2,3,4], which comprises over 90% of thyroid cancers. Recurrent or persistent diseases can manifest themselves as a biochemical disease without structural evidence, nodal metastasis, remnant disease, and distant metastasis. Likewise, surgery is considered the treatment of choice for most resectable recurrent neck diseases. However, an understanding of the biology of recurrence has led to recommendations for the surveillance of small-volume disease. In addition, a range of novel therapeutic options has emerged, and these are now under intense study and offer an alternative to surgery in case of recurrence. An understanding of the individualization of these options is critical for managing the clinical team.

Although associated with disease-specific survival rates of nearly 100% at 5 years, 10–15% of the patients with WDTC have locally advanced disease. This patient group is at high risk of locoregional recurrence in addition to distant metastases often associated with more limited survival [5]. Although disease presenting in an aggressive manner will tend to recur within the first 5 years, relapses remain a significant issue for long-term surveillance. More than 30% of patients with WDTC recurrence are diagnosed after the first decade of follow-up, and the most common sites of involvement are cervical lymph nodes [6]. Among the subtypes of WDTC, papillary thyroid carcinoma is the major source of locoregional tumor recurrence occurring in the central or lateral neck compartments.

Disease recurrence rates include a continuum that ranges from less than 1% in patients with the lowest risk to over 50% in those with high risk, according to the American Thyroid Association (ATA) Initial Risk Stratification System [1]. In a recent series, 75 out of a cohort of 685 (11%) WDTC patients developed cervical structural recurrence, 36 (5.2%) developed distant metastases (of which 2.8% with both local and distant disease), and 14% died from their disease between 26 and 156 months from initial diagnosis (mean of 6 years) [7]. Rates of persistent/recurrent disease were 3.6%, 13.6%, and 71.4% for low-, intermediate-, and high-risk patients, respectively. At the end of the follow-up periods, which ranged from 4 to 243 months (mean: 71.5 months), 0.3% of patients in the low-risk group, 0.9% in the intermediate-risk group, and 15.2% in the high-risk group had died of their disease [7]. In contrast to those patients with locoregionally advanced disease, occult lymph node metastases have no impact on survival rates and have minimal influence on relapse [8].

The patients at greatest risk of death from the disease are older at the time of diagnosis, presenting with distant metastatic disease, or having a gross extra thyroidal extension to adjacent anatomical structures. Age is a risk variable for mortality included in the AJCC-UICC TNM staging system, where a cut-off point, defined as 55 years at diagnosis, guides the classification criteria [9,10]. However, the effect of age progression on prognostic impairment is continuous, as its influence is noted as early as 30 years of age at diagnosis [11]. The TNM classification illustrates the risk of death from the disease [9], whereas the ATA risk classification stratifies patients in terms of risk of disease recurrence, which does not always translate into an increased risk of death [1]. In the ATA risk classification, risk factors for recurrence include tumor extension (extrathyroidal or extranodal) to adjacent structures or gross residual disease after resection, palpable cervical metastases, extensive vascular invasion, and inappropriately elevated serum thyroglobulin titers postoperatively [1]. Beyond initial treatment, based on the initial response to therapy, dynamic and ongoing risk stratification is also advised [12]. Therefore, careful analysis of patient and tumor-related factors is required to individualize therapy which also includes additional treatment or active surveillance, intensity and modalities of the therapies, and follow-up recommendations. Factors including operability, disease trajectory, availability of capable surgeons, and overall prognosis, as well as the functional impact of major ablative surgery and the overall status of the patient, must be balanced when considering the suitability of surgery versus other therapeutic options. In this way, clinical teams can optimize the long-term prognosis for these patients.

## 2. Methods

In the preparation of the present review, the guidelines from the American Thyroid Association (ATA) [1], the National Comprehensive Cancer Network (NCCN) [2], the European Society for Medical Oncology (ESMO) [3], the European Thyroid Association [13,14,15], and the Japan Associations of Endocrine Surgeons were included as references [4]. The present review was based on updates of the existing guidelines and studies published, with relevant cohorts, to provide robust evidence. The meaningful literature was considered through a comprehensive PubMed search.

The ATA and the NCCN recommendations are renowned North American guidelines, while the ESMO and the ETA guidelines are relevant European references, as the Asian continent was represented by the Japan Associations of Endocrine Surgeons guidelines in an attempt at global representation. This rationale guided the choice of these references for the composition of this review.

## 3. Recommended Diagnostic Work-Up

Clinical management after initial surgical treatment of WDTC should include periodic physical examinations, serum thyroglobulin (ideally using serial assessment with the same assay) and serum anti-thyroglobulin antibody levels, and structural assessment with high-resolution cervical ultrasonography, preferably with color Doppler. The frequency of these assessments should be considered in a risk-adapted manner based on the risk for recurrence or persistent disease, as well the observed response to the initial therapy [1]. Both serum thyroglobulin levels measured by ultrasensitive methods and high-resolution ultrasonography are sensitive for detecting persistent/recurrent disease following total thyroidectomy and radioactive iodine (RAI), potentially facilitating the detection of small-volume disease [1,2,3,16].

An elevation of serum thyroglobulin level should be followed by high-resolution ultrasonography of the neck. If no evidence of structural cervical disease is detected, the next step is to obtain computed tomography (CT) of the neck, mediastinum, and lungs [16]. Serum thyroglobulin levels correlate not only with tumor volume but also predict the lesion location. Levels less than 10 ng/mL suggest nodal disease, while levels up to 500 ng/mL may indicate pulmonary metastases, and levels over 1000 ng/mL bone metastases [17]. However, this marker loses its accuracy if there are detectable serum anti-thyroglobulin antibodies, which themselves can be used as a potential marker of disease recurrence. Thyroglobulin doubling time also helps the assessment of tumor progression, although the data are not sufficient to use this marker for precise prognostication. A doubling time of less than one year may be a poor prognostic factor [17]. However, less differentiated variants of papillary thyroid carcinoma may show low thyroglobulin levels. Serum thyroglobulin may also rise due to thyroid bed persistent benign remnant tissue in patients who have not received radioactive iodine ablation [1], but typically at a rate much slower than with structural disease manifestation and progression. Distinguishing the origin of this marker—from normal thyroid tissue or tumor progression—may be difficult in this context [1].

If suspicious lesions are detected in the neck, and the goal is for treatment, cytological diagnosis by fine needle aspiration is indicated, although some foci of the suspected disease may not be amenable to this procedure due to its anatomic location. In these cases, surgery may still be considered [13]. Thyroglobulin measurement in the needle wash-out complements the cytological analysis and helps confirm the diagnosis [1,2]. For surgical planning, cross-sectional images such as CT or magnetic resonance imaging (MRI) are useful to evaluate disease extent and possible extra thyroidal or extra nodal extension and to help understand the extent of previous surgery and whether compartment dissection or focused removal is warranted [2].

Before reoperation in the central compartment, a laryngoscopy to assess the functional status of vocal folds is recommended. If invasion of the aerodigestive tract is suspected, in addition to cross-sectional images obtained by CT or MRI, esophagoscopy and tracheo-bronchoscopy may be warranted to rule out tumor extension to the esophageal and tracheal lumen to select the appropriate treatment options [3].

If there is suggestion of progressive disease, total body radioactive iodine scanning could be indicated to help therapeutic decisions based on RAI avidity and possible treatment of that disease if no further surgery is recommended, following NCCN recommendations [2]. However, the last ETA Consensus Statement for the indications for postoperative RAI ablation underlined that RAI whole-body scanning has a low impact on therapeutic decision making and may be associated with stunning thyroid remnant tissue and metastases [15]. In addition, [^18^F]2-fluoro-2-deoxy-D-glucose positron emission tomography (^18^FDG-PET), although not a routine imaging study in WDTC follow-up, is indicated if a rapid rise in serum thyroglobulin level occurs with no structural disease identified on RAI scans and conventional axial neck imaging. ^18^FDG uptake in that situation raises the suspicion of tumor dedifferentiation [16], as this can be associated with a worse prognosis since such tumors are usually refractory to RAI treatment [3]. ^18^FDG-PET combined with CT with contrast is useful for assessing the extent and precise topography of the lesion [3].

## 4. The Basis for Therapeutic Decision Making

The clinical question addressed in this review is what management options exist for patients with locoregional recurrent WDTC in the neck following initial treatment. Initially, it is important to distinguish the local recurrence in the thyroid bed or in the residual glandular tissue from the recurrence in the regional lymph nodes of the central or lateral compartments of the neck [13]. Additionally, one should contemplate concerns about several relevant variables, as shown in Table 1. This process starts from complex and challenging scenarios when approaching the patient with incomplete disease response, and discussion in a multidisciplinary and thyroid expert tumor board helps in complex decision-making conducting.

Locoregional control is an important endpoint as tumor progression can be associated with significant morbidity and affects the quality of life because of the proximity of critical structures such as the trachea, the larynx and its innervation, the esophagus, and the great vessels of the neck and mediastinum. In the context of palliation, the aim is to reduce tumor burden and to prevent the progression of the disease, especially the involvement of these vital structures [1].

In clinical practice, there are some patients with minimal symptoms in the presence of advanced locoregional recurrence. While dysphonia can be a relatively early complaint, dyspnea, stridor, hemoptysis, and dysphagia usually present later as more troubling symptoms heralding the involvement of large cervical vessels or adjacent neural structures, such as the vagus and phrenic nerves. Lesions can present as fixed cervical masses, with pain and ulceration generally presenting as late events. It is particularly difficult to recommend potentially highly invasive treatments in patients with minimal symptoms [1,5,18].

In many cases, prolonged survival is observed even in the presence of active disease. A balance is therefore sought between the probability of cure, preventing local disease progression in critical areas, and the quality of life associated with the disease versus potential treatment. The available evidence on treatment efficacy is mostly obtained from retrospective data, complicating the balance between the effect of therapy versus the natural history of the disease [3].

There is no evidence that patients with rising thyroglobulin levels without identifiable progression of structural disease (biochemical disease) have a high future risk of disease-related morbidity or a decrease in disease-specific survival [1]. The same is true for elevated anti-thyroglobulin antibody titers. Therefore, additional therapies are not recommended based solely on increasing levels of tumor markers or an incomplete biochemical response. In such cases, TSH suppressive therapy using levothyroxine and appropriate serial imaging is considered the first option because it is unclear whether RAI therapy improves survival in these situations, considering also that frequently these patients no longer demonstrate RAI avidity [19].

Most institutions follow the therapeutic sequence recommended by evidence-based guidelines, which may suggest surgical resection observing oncologic principles in curable patients; ^131^I therapy for RAI responsive disease; external beam radiation therapy, or other local treatment such as ablation, for progressive local recurrences not eligible for surgery; and systemic therapy with kinase inhibitors in those with iodine refractory and progressing tumors, especially in the setting of systemic disease. However, the guidelines do not address controversial areas where existing evidence is limited [13].

Suppressive TSH therapy is recommended for patients without clinical contraindications. In patients with structurally identifiable relapses, decreased rates of disease progression, as well as reduced recurrence and death rates, have been observed with TSH suppression, although it is not clear what is the exact appropriate level of suppression for each situation. According to the ATA guidelines, high-risk patients should receive suppressive doses of levothyroxine to maintain serum TSH levels below 0.1 mIU/mL [3].

Figure 1 shows an algorithm that summarizes this review’s recommendations for the management of recurrent neck disease in patients with WDTC.

## 5. Current Treatment Modalities

### 5.1. Surgery

Surgery is the usual treatment of choice for biopsy-proven cervical persistent or recurrent WDTC [1,2,3,4,18]. Upon its diagnosis, the patient should undergo careful imaging of the central and lateral compartments of the neck, as well as the mediastinum. Following a comprehensive assessment, a planned neck dissection including the affected nodal compartment: central (levels VI and VII) and/or lateral (levels II–V) sparing uninvolved critical anatomical structures should be considered. If the patient previously had a comprehensive neck dissection, the goal is an adequate removal of the lesion [1,2,20].

The ATA guidelines recommend surgery for central neck compartment nodes ≥ 8 mm or lateral neck nodes ≥ 10 mm in their short-axis diameter [1]. However, small nodes (less than 2 cm) can often be actively monitored for years before requiring surgical intervention, particularly in favorable locations in the lateral neck. Nonetheless, prospective and randomized trials conducted for periods longer than a decade and evaluating the outcome of early surgery versus active surveillance are lacking, and it is not available the best level of evidence upon which to base clinical decisions [13].

A retrospective Canadian study included 1062 WDTC patients submitted to surgical treatment with a mean follow-up period of 4.1 years. Locoregional recurrence was observed in 11% of the cohort, and almost 50% of this subgroup was submitted to surgery (and sequential RAI treatment in around half of them). After reoperation, the absence of structural disease was obtained in 67% of the patients following multimodal therapy. Five-year overall survival was 96% for those without new relapses, 95% for those with further locoregional recurrence, and 72% for those who developed distant metastases (3% out of the study population). Of the patients who developed distant metastases, 51% had a prior locoregional relapse [21].

A 12% rate of cervical nodal recurrence was reported in previously dissected lateral compartments of the neck in a Japanese series (744 patients, 113 months of mean follow-up time), while 7% presented with relapses in the contralateral side of the neck. Extrathyroidal extension was a relevant predictor for both outcomes [22]. In another series that also included lateral neck dissection (307 patients submitted to 429 surgical procedures for cytopathology-confirmed neck relapses), overall regional control at 10 years was 88%, observing a worse performance in younger patients (less than 24-year-old at diagnosis), nonetheless, a superior disease-specific and overall survival in patients under 50 years-of-age was noted. In-field lateral neck control was 96% at 10 years, and cervical levels III and IV were the most affected by recurrence [23].

When recurrent tumor extends to anatomical structures beyond lymph nodes, the objective of the surgical approach should be the resection of all gross tumors with preservation of uninvolved functioning and vital structures as best as possible. Adjuvant postoperative radioactive iodine is indicated if the tumor retains avidity and depends on the dose of radioiodine the patient may have received previously, as discussed later. Reoperations entail greater technical difficulty and risks than primary procedures, particularly if the recurrent disease presents in a site of previous surgical intervention, in comparison with “out of field” recurrences [1,24].

The difficulty and morbidity of reinterventions are related to the anatomy of the operated region, the degree of fibrosis and scarring, the extent of recurrence, and the surgeon’s experience. The main risks of reoperating on the central compartment of the neck include injury to the recurrent laryngeal nerves and parathyroid glands, although some experts have performed it without increased risk compared to primary surgery. Studies have shown varying complication rates. The reported rate of transient vocal fold paralysis ranges from 0% to 20%, and the rate of permanent paralysis ranges from 0% to 12%. The rate of permanent hypoparathyroidism after central compartment node dissection ranges from 0% to 15%, with a higher incidence of temporary hypoparathyroidism [13,23,25,26,27]. Recurrence, manifested as soft tissue disease, frequently occurs in the area of Berry’s ligament and cricothyroid joint, where there is close anatomical proximity between recurrent laryngeal nerve and the thyroid tissue; consequently, some glandular remnants may have been left behind at the time of initial thyroidectomy, which reduces nerve manipulation but implies increased risk of trauma to the neural structure at the reoperation compared to the nodal metastases alone [26,28].

The role of surgery in the optimal treatment of gross recurrent disease must be balanced against the surgical risks. Extended resections can range from removing structures that can be safely excised without major consequences, such as the cervical strap muscles, to highly functional structures, such as the larynx. On the other hand, radical resections may be required leading to permanent sequelae in patients with limited symptoms but whose tumors, if left untreated, would progress toward unresectability. An example is total laryngectomy in patients with a tumor invading the laryngeal lumen that can cause symptoms limited to mild dysphonia or dyspnea. It seems reasonable to limit these resections to patients with advanced and localized tumors without distant metastasis and, therefore, a potentially curative treatment aim.

The recurrent laryngeal nerve can be invaded both through direct tumor extension and through its metastases. Management of the invaded recurrent laryngeal nerve depends in part on the functional status of the ipsilateral and contralateral vocal fold, the relationship of the tumor to the nerve (adherent versus encasing), and the overall staging of the disease (presence of distant metastases and the possibility of being able to remove all macroscopic tumor). Usually, if there is evidence of invasion of the recurrent laryngeal nerve by the tumor, but the neural function is maintained, the structure should be preserved by shaving off the invading tumor [28]. In a Japanese series, 83% (15/18) of the patients who underwent partial resection of recurrent laryngeal nerve layer, defined by the preservation of less than 50% of its original thickness, achieved functional vocal folds or near-normal phonation 1 year after surgery [29]. However, if neural preservation implies leaving gross tumor behind, one should consider a sacrifice, even if it had some function preoperatively, but ensuring that the contralateral nerve is functioning and not directly involved by the tumor, as no gross disease is left in other areas of the neck making resection of the nerve useful in achieving the eradication of all visible tumor. In turn, laryngeal recurrent nerve resection is recommended if the patient already has dysphonia and impaired laryngeal function [4].

A recent retrospective study included 65 patients with documented recurrent laryngeal nerve invasion by thyroid carcinoma, its metastases, or both. However, only 39% had preoperative voice complaints, and preoperative vocal fold paralyses were documented in 43.5% of the patients. The nerve was resected in 74% of the cases, and the rate of vocal fold palsy was 82% after 6 months after the surgery. Interestingly, it was not possible to identify risk factors associated with postoperative vocal fold palsy in the patients with no preoperative dysfunction, as the resection of the invaded nerve or the complete resection of the tumor from the nerve were not associated with statistically significant improvement in survival. Locoregional recurrences were observed in 64% of the patients during the follow-up time [30].

Therefore, careful preoperative patient counseling is mandatory, but intraoperative decision making may be required, particularly if there is concern about the function of both nerves. In such situations, intraoperative nerve monitoring may be useful in making decisions about nerve management [28]. In cases of segmental resection, an immediate intraoperative repair of the recurrent laryngeal nerve includes end-to-end neurorrhaphy, free nerve graft anastomosis, or recurrent laryngeal nerve anastomosis to regional nerves such as ansa cervicalis and vagus [31]. The recurrent laryngeal nerve has different branches for adduction and abduction functions. Its reinnervation nerve may restore the vocal fold tone, but it will hardly restore both abduction and adduction functions.

Aerodigestive tract invasion by WDTC is infrequent but has a negative impact on survival, especially when there is intraluminal involvement. When there is evidence of tracheal invasion, superficial shaving, window resection, and segmental tracheal resection with end-to-end anastomosis are the most used surgical methods, and specific indications will depend on the extent of involvement [32]. Segmental tracheal resection is associated with better local control when disease affects the tracheal mucosa. Although it is often feasible, accurate knowledge of local anatomy and surgical technique are vital, as is a proper patient selection [33]. Evidence of invasion of the esophageal wall through its mucosa is rare, and surgical treatment is associated with a high degree of morbidity. However, the surgical treatment of invasion restricted to its muscular layer, preserving the mucosa, is more straightforward [34]. Therefore, careful preoperative endoscopic assessment, including inspection of the esophageal mucosa, is recommended.

In a series of 69 WDTC patients with aerodigestive tract invasion, including both primary tumors and recurrences, older individuals were overrepresented (one-third were older than 65 years). Interestingly, only 10% of papillary thyroid cancers were classified as aggressive subtypes. Thirty-four patients (49%) underwent organ-sparing resection of the trachea or larynx, and esophageal resection was performed in 56% of the patients. Thirty-three patients (48%) required a tracheostomy at the time of surgery, 27% were permanently tracheostomy dependent, and the incidence of surgical morbidity was 31%. However, 71% of the patients were able to eat normally, 59% had intact speech, and 86% were free of chronic pain. Although regional recurrence was diagnosed in 15% of the patients, 23% developed distant metastases with a median follow-up of 58 months. The authors did not find any statistically significant difference in overall or disease-free survival between patients submitted to resections with negative margins over resections with microscopically positive margins or even gross disease, but most patients received adjuvant external beam radiotherapy [34].

### 5.2. Active Surveillance

When there is suspicion of recurrent disease through highly sensitive serum thyroglobulin assays, localization tools such as neck ultrasonography and cross-sectional imaging modalities can identify even very small volume disease [1]. The ATA guidelines state that neck lymph nodes with up to 8 mm in their smallest diameter in the central compartment and up to 10 mm in their smallest diameter in the lateral compartment may be suitable for active surveillance [1]. If this approach is chosen, cytological confirmation of the disease can be deferred, although serial imaging exams and periodic assessment of serum markers should be performed. The NCCN guidelines also consider active surveillance appropriate for patients with the stable low-volume disease and distant from critical structures [2]. Active surveillance should be conducted in properly selected patients by serial high-definition ultrasonography and/or cross-sectional imaging, serum thyroglobulin, and serum anti-thyroglobulin antibody monitoring [1,13].

A retrospective study from Memorial Sloan-Kettering Cancer Center using active surveillance in 191 patients observed a slow progression of thyroid bed nodules ≤ 11 mm (median size of 5 mm). Favorable conditions for this outcome included the absence of additional abnormal cervical lymph nodes beyond the index lesion, a lack of suspicious ultrasonographic features, and stable serum thyroglobulin. When these features were present, thyroid bed nodular disease was reported to increase at a rate of only 4% over 5 years of median follow-up. The late appearance of a cervical recurrence following the initial surgery also suggests a favorable condition indicating an indolent, slow-growing tumor. However, in this ultrasonographic study, lesions did not have cytological confirmation of disease and, therefore, may have included some benign nodules [35].

The proximity of the tumor to functional and vital structures, such as the recurrent laryngeal nerves, must be considered since tumor infiltration can occur independent of the lesion dimension. Additionally, some primary tumor characteristics should be recognized, notably high-grade histology, rapid thyroglobulin doubling time, lack of RAI avidity, increased ^18^FDG-PET avidity, and molecular markers associated with aggressive behavior [1]. It is important that patients understand this process and remain committed and comfortable with the recommended follow-up. It is assumed that the risk of treatment should not exceed the risk of the disease, and considerations about comorbidities and life expectancy must be individually considered.

### 5.3. Radioactive Iodine

Adjuvant RAI therapy is indicated for ATA high-risk patients and for selected intermediate-risk patients in an attempt to improve disease-free survival and specific mortality [1,2,3,4,15,36]. Surgical and pathological reports, such as post-operatory (between two weeks and two months) serum thyroglobulin measurements and neck ultrasonography, should also be considered [15]. Post-therapy radioactive iodine whole-body scan is recommended to define the extent of thyroid remnant and may detect the presence of unsuspected disease [15].

According to the ATA guidelines, recurrent regional lymph node metastases diagnosed on iodine whole-body scanning can be treated with RAI in cases of low-volume disease or in combination with surgical resection, although surgery is preferred, especially for lesions larger than 1 cm. RAI can also be employed with adjuvant intent following surgical resection of aerodigestive invasion in the presence of residual iodine-avid disease [1]. Tumor volume should always be considered when faced with such a therapeutic attempt.

The maximum tolerated absorbed dose should also be taken into account for each case. Repeating RAI administrations after a cumulative activity of 600 mCi should be on a per-patient basis [3] since a cure is unlikely in this situation. Tumor burden, RAI avidity, and response to previous treatments should be considered in decisions regarding continuing RAI therapy [37]. Patients with high (>600 mCi) cumulative lifetime doses are, in turn, at increased risks of myelosuppression, second primary cancers, and other long-term toxicities and should be appropriately monitored [2].

The use of this therapeutic modality requires high serum TSH levels and a diet with low iodine content. TSH stimulation can be achieved with levothyroxine withdrawal for 4 or 5 weeks leading to an intrinsic TSH elevation, or injections of human recombinant TSH (rhTSH) may be administered. Although expensive, it has benefits in terms of patients’ quality of life and is increasingly recommended in modern practice. Similar performances with both modalities of TSH stimulation are observed [4,15]. Although the ATA, ETA, and NCCN guidelines still recommend levothyroxine withdrawal for the treatment of distant metastases treatment [1,2,13], rhTSH is now becoming the standard for many indications. All forms of RAI treatment should be postponed for at least 6 weeks after administration of any iodinated contrast medium [3].

Many patients with recurrent and/or metastatic tumors no longer concentrate radioactive iodine and are considered RAI-refractory, although publications differ in their criteria for the absence of RAI uptake on scintigraphy and in the definition of tumor progression after RAI therapy [13]. In summary, these are lesions that do not have or lose the ability to concentrate iodine, being also possible a mixed pattern of iodine accumulation or progress after RAI administration despite iodine avidity at the time of initial treatment [13,19]. Following the European Thyroid Association recommendations, the decision to continue or not with RAI treatment in a patient with RAI-refractory lesions, as well as in patients who have received multiple treatments with RAI, should be addressed by a multidisciplinary team, weighing the benefits and disadvantages of continuing RAI and considering the combination with local treatments [13].

For re-staging purposes, ^18^FDG positron emission tomography is the first-line isotopic imaging technique for patients with RAI-refractory disease [3], and its uptake is associated with a worse prognosis [13,38]. Currently, clinical trials are under development trying to validate some selective tyrosine-kinase inhibitors as drugs to restore iodine uptake and also considering the tumor genetic analyses.

### 5.4. External Beam Radiotherapy

In highly selected differentiated thyroid carcinoma patients, external beam radiotherapy (EBRT) can be considered as a postoperative adjuvant therapy (for treatment of presumed residual disease and low likelihood response to RAI) or as definitive treatment of unresectable relapses or for palliation (symptom control) [39]. However, it should be emphasized that neck EBRT for WDTC is extremely rare, and EBRT is not considered unless the patient has been evaluated by a multidisciplinary team with significant experience in treating advanced thyroid cancer. It is all too common in practice to see patients who have been inappropriately treated with EBRT for advanced WDTC, which can significantly limit their future treatment options, particularly surgery.

While there is no evidence to support the indication of EBRT after initial adequate surgical therapy, even in the setting of aggressive tumor or other variables of risk [1], it is a modality that can be considered if there is evidence of gross residual disease not amenable to radical surgery or RAI, especially if further surgery by an experienced thyroid surgeon is not an option and it is not possible without significant morbidity [1,2,3,39]. Patients with minor soft tissue invasion, such as sole recurrent laryngeal nerve involvement, should not routinely undergo EBRT [25]. EBRT should also be avoided in young patients or patients with iodine avid tumors [39,40].

Publications have described different results considering retrospective restricted cohorts that included patients with different clinical situations and tumor types. Although WDTC has a low sensitivity to EBRT [4], better disease control was observed in patients with tracheal invasion and/or esophageal invasion who have been treated with aggressive surgical resection followed by RAI plus EBRT. No difference, however, was found in overall survival or distant recurrence-free survival when that group was compared with those treated with surgery followed by RAI [25]. A meta-analysis that included nine studies that considered RAI and EBRT after surgery concluded that EBRT improved locoregional recurrence-free survival but had no impact on overall survival in patients at increased risk of relapses, such as those of advanced age, locoregionally advanced disease, macro or microscopic residual tumor, and visceral invasion [41]. However, these risk variables are not EBRT indications per se. The patient context should be individually analyzed.

Although studies have considered its acute toxicity to be tolerable [6], the late adverse effects of EBRT should be considered in the face of a tumor usually compatible with prolonged survival, even in the presence of its progression. In addition to the acute effects on skin and mucosa, there is impairment of salivation and swallowing, as a possibility of stenosis in the esophagus and trachea, especially in the long term, with significant impairment in quality of life. Additionally, EBRT significantly limits future surgical approaches, and it is reserved for patients in whom future surgical intervention is not feasible [25].

EBRT can also be used in the palliative setting [1] when the disease or the patient condition precludes surgery, although the expectation should not be of complete response but an attempt to control local disease progression. While for curative intent, doses of 60 to 70 Gy are indicated, in the setting of palliation, lower doses should be recommended [42]. The role of EBRT remains controversial, and its use is individualized [25,39,41]. Carefully weighing the risk of disease progression and death with the potential benefits and risks of additional therapies promotes an individualized, risk-adapted management and follow-up plan [42].

### 5.5. Ultrasonography-Guided Percutaneous Ablation

Ultrasonography-guided percutaneous ablation is a localized treatment modality [1,2]. It can be considered for patients with localized lymph node metastatic disease. Ideally, small-volume (<2 cm) lesions that are low in number (<4) are selected in patients who could undergo surgery but are poor surgical candidates (such as those having considerable risk morbidity in a region already submitted to other therapeutic modalities, severe clinical comorbidities or patients who refuse additional surgery). It is considered a nonsurgical form of focused treatment, and previous treatment modalities should be taken into consideration [1,14].

The treatment should be performed before the initiation of any systemic treatment and may postpone this indication [1]. However, in the presence of a progressive single lesion, the local treatment can be conducted without temporarily interrupting the systemic therapy or withdrawing it just for a few days [13].

Different forms of ablation range from thermal (as radiofrequency, laser, microwave, and cryotherapy) to chemoablation (as ethanol). Studies have described reasonable results but with short-term follow-up and small cohorts [14,43,44,45,46,47,48]. These thermal techniques achieve partial or complete necrosis of the neoplastic tissue through an increase (more than 55 °C in radiofrequency, for instance) or decrease (from −40 °C to −140 °C through cryoablation, for instance) of the intra-tumoral temperature [14,15]. On the other hand, the choice of a specific technique should be based on the personal competences and resources of the centers [14].

Reasonable rates of index lesion regression and biochemical responses have been reported. A study conducted with 63 patients submitted to ethanol injection treatment of locoregional metastases reported a rate of 84% of complete response, although in a mean follow-up time of only 38.4 months [46]. Another report observed no progression of locoregional metastasis treated with percutaneous radiofrequency (21 lesions), while it was diagnosed in 5 of 21 lesions submitted to ethanol injection between 4 and 11 months after the therapy [47]. Meanwhile, a meta-analysis gathered 10 publications and 415 metastatic lesions treated with radiofrequency and ethanol injection, and, again, the success rate with radiofrequency was higher but with no statistically significant difference (69% and 53%, respectively). However, the reduction in serum thyroglobulin was superior in the patients treated with ethanol injection. The rate of complications was 1.6% for both techniques [48].

As with EBRT, it is likely that ablative approaches to recurrent disease will have a role to play in the control of specific lesions but are less likely to impact overall survival. However, although considered minimally invasive, these procedures are not free of local risks due to the possibility of injury to critical structures adjacent to the neoplastic lesion. Local pain, skin burns, recurrent laryngeal nerve injury, and dysphonia are related effects, but the reported studies did not observe any major complications [14,43,44,45,46,47,48]. In thermal ablation, hydrodissection—the injection of saline or glucose solution around the target lesion—should be performed to provide a secure distance between the lesion and surrounding structures [14].

The patient should preferably be under sedation and in a hospital environment. More than one session may be required for local control, especially for larger lesions. Response monitoring is conducted through serial ultrasonography exams that analyze the variation in tumor volume and the pattern of vascular flow (color doppler or contrast), combined with serial measurements of serum thyroglobulin and anti-thyroglobulin antibody titers. Importantly, it can be observed an increase in the volume of the target lesion after thermal ablation, as it can occur a rapid increase in serum thyroglobulin during the first 48 h after the therapy, which should not be interpreted as disease progression [14].

This approach should preferably be used in the context of a clinical trial when there is no longer any intention of reoperation. The specific local effects of ablation on an eventual surgical re-approach are unknown.

### 5.6. Systemic Therapy

Although local treatments are preferred, systemic therapy may be considered for disease and symptom control in WDTC patients with neck recurrences, preferably if the disease is advanced, multimetastatic, and progressive [49]. No single imaging modality provides complete information on tumor biology and its dynamics of growth. A combination of imaging techniques, as well as other diagnostic tools, is indicated to define which tumor lesion requires systemic treatment [13]. In turn, also relevant for the indication of the other therapeutic resources mentioned, the participation of an experienced multidisciplinary team is fundamental if systemic treatment is considered, especially for defining when to start.

Doxorubicin may participate as a radiation sensitizer and can be considered in association with EBRT in patients with locally advanced disease [1]. However, the results of conventional cytotoxic therapy, with single agents or combinations in RAI-refractory tumors, are disappointing, and there is no role for its routinary use [3,4,49]. For now, targeted therapies able to inhibit abnormally activated tyrosine kinases are the systemic treatment used as the first line in metastatic and progressive WDTC [49].

Tyrosine kinase inhibitors are oral multi-target cytostatic drugs that have been approved and tested for patients with relapse or persistence and refractory to the other therapeutic modalities mentioned. They can be associated with improved progression-free survival, but they are not curative [2]. Lenvatinib and sorafenib are the standard first-line systemic therapy for unresectable and RAI-refractory lesions. Clinical applications include rapidly progressive (significant structural disease progression during the last 6–12 months) [2] disease with multiple and RAI-refractory lesions, symptomatic conditions with also multiple and RAI-refractory lesions [1,2,3], or life-threatening tumors [1,2]. As angiogenesis inhibitors, the risk of delayed wound healing and bleeding should be considered. The same reasoning applies to those with tumor invasion of vessels and skin [4]. In the asymptomatic patient, the timing for starting the therapy is also a dilemma given the possible side effects, although postponing administration may be problematic. The pace of disease progression is an important tool for treatment decisions [2].

The RECIST (Response Evaluation Criteria in Solid Tumors) concept [50] is used for defining target lesions and quantifying the response to therapy, although it should be considered with caution since a radiological progressive lesion can be clinically irrelevant, while other, even with a minor increase in volume, can have a relevant clinical impact due to its anatomical position [13]. Sorafenib was considered eligible for the treatment of iodine-refractory locally advanced or metastatic WDTC in a clinical trial (DECISION trial) [51] that observed a significantly higher rate of progression-free survival among patients submitted to the treatment versus those in the placebo arm. While objective response rate was 0.5% in the control group, it was 12%, albeit partial, in those receiving sorafenib. The average duration of responses was just over 10 months. Disease control (partial response or stability lasting ≥ 6 months) was more frequent among the patients who received the drug. Overall survival was similar in both groups, however.

The SELECT trial [52] investigated the use of lenvatinib in iodine-refractory advanced disease observing reasonable responses, even superior to that observed in the patients treated with sorafenib, although the cohorts were not comparable. Overall survival was also similar in both arms but superior in patients older than 65 years treated with lenvatinib. Lung and lymph node lesions had better responses than liver and bone metastases. Lenvatinib is the first preferred systemic regimen [2]. RAI-refractory lesions included, in the commented trials, the presence of at least one target lesion without uptake, a tumor that had uptake but progressed within 12–16 months of treatment, a tumor that progressed after two RAI treatments within 16 months of each other, or patients who received a total RAI dose of higher than 600 mCi [2,51,52]. A first imaging evaluation should be performed to verify the effectiveness of the systemic therapy 2 or 3 months after the start of its administration [13].

On the other hand, there is the prospect of using these drugs as neoadjuvant therapy in patients with unresectable tumors or borderline resectable disease, which is not thought amenable to surgical resection without considerable morbidity. Some patients may benefit from this approach, as related in a few individual cases where its use provided tumor bulk reduction followed by surgical resection with curative intent and without the sacrifice of vital structures [53]. However, it is not yet known how to select the group of individuals who would respond to this therapeutic attempt. This is ideally based on an understanding of the mutational analysis of the tumor, which potentially allows targeted therapy on an individual level. Consideration of the potential to avoid an incomplete surgical resection with neoadjuvant therapy should be balanced against the potential of non-responders to further progress to truly unresectable disease. Careful assessment of potential areas where there is an “R2 interface” may allow patient selection for this treatment approach. However, little is known of the long-term outcomes, and clinical trials are underway to better assess patient and tumor factors that might guide therapeutic decisions in the future.

Before starting systemic therapy, it is recommended a careful analysis of the clinical features of the patient as well the number, the size, the site, and the rate of growth of the lesions. Whenever possible, a local treatment should be preferred, and systemic therapy should be postponed until evidence of a multimetastatic disease with rapid growth, according to RECIST, since a negative impact of these drugs on the quality of life is a possibility [13]. There are clinical contraindications to targeted therapy, and the doses must be adjusted or even interrupted considering its toxicity, which is more frequent in older patients, which will clearly affect the desired effect. Treatments, in turn, should be maintained as long as they are effective (usually for about 12 to 24 months) [2], which may even include reducing the rate of tumor growth [13] tolerated and accepted by the patients. On the other hand, for local tumor progression, locoregional treatments (such as surgery, EBRT, and ablation) can be offered to the patient without discontinuing the systemic therapy or interrupting it in order to optimize disease control [13]. Associated side effects may have a significant impact on quality of life [2] and are related to “off-targeted” activities [49]. Major adverse events include hand–foot syndrome, diarrhea, exanthema, loss of weight, systemic arterial hypertension, fatigue, and others, particularly at the beginning of the therapy [1].

A second generation of tyrosine kinase inhibitors has been approved. They are specific for a specifically altered oncogene with greater tolerability [49]. Selpercatinib and pralsetinib are highly potent RET-selective protein tyrosine kinase inhibitors of both point mutations and fusions and with a marked and more durable anti-tumor activity. They received 2020 the United States Food and Drug Administration (FDA) approval for treating metastatic RET fusion-positive papillary thyroid carcinoma, among other tumors such as medullary thyroid carcinoma [54].

NTRK (tropomyosin receptor kinase) fusions are rare but can be found in several solid tumors, including thyroid papillary carcinoma, and drugs (such as larotrectinib, a highly selective inhibitor of TRK) against solid cancers expressing the TRK gene fusion have been approved [49,55], while other systemic modalities are also being studied, as the possibility of association of antiangiogenetic agents plus immune checkpoint inhibitors [3]. For now, several clinical trials enroll kinase inhibitors, *BRAF* V600E mutant inhibitors, TRK inhibitors, *RET* inhibitors, and anti-PD-1 antibodies [2]. Since they are not curative, other therapeutic strategies are being explored. In addition to the development of new drugs, characterizing resistance mechanisms and their translation to the clinic are critical for improving patient outcomes [54].

## 6. Conclusions

Surgical resection, either as primary or salvage therapy, is the treatment of choice for patients with WDTC. Recurrences manifest themselves mainly as metastases to cervical lymph nodes, particularly from papillary thyroid carcinoma. It is challenging, in turn, to identify and eradicate the nodal disease in a previously operated neck, supporting the notion that an appropriate neck dissection is an ideal treatment for node involvement both initially and at the time of reoperation.

When the patient carries a tumor refractory to standard therapies of choice (surgical resection followed by RAI therapy), other approaches should be individually considered. An attempt is made to prevent the progression of disease threatening the vital structures of the central neck, and an organ-sparing surgical approach seems to represent the best balance between locoregional control, functional, and quality of life outcomes. Although ideally, the achievement of negative resection margins is sought, realistically, this is limited by the extent of tumor involvement, the proximity to critical structures, patient comorbidities, and the presence of distant metastases. It is possible to consider the options of RAI therapy, active surveillance, external beam radiotherapy, ultrasound-guided percutaneous ablation, and systemic therapy, with the possibility of using more than one modality, synchronous or sequentially.

The ideal approach includes balancing the quality-of-life impact that the disease has at present and will have in the future against the impact of treatment for patients who, in general, have prolonged survival, even in the presence of advanced locoregional and distant disease. Such balanced treatment decisions are best made in a multidisciplinary setting with the input of clinicians from a broad range of specialties while respecting the views of the individual patient concerned in a shared decision context. Importantly, thyroid surgeons must be familiar with the natural history of this disease, the proper diagnostic workup for each situation, and indications and techniques of comprehensive and compartmental, oncological safe neck dissections.

## Figures and Tables

**Figure 1 cancers-15-00923-f001:**
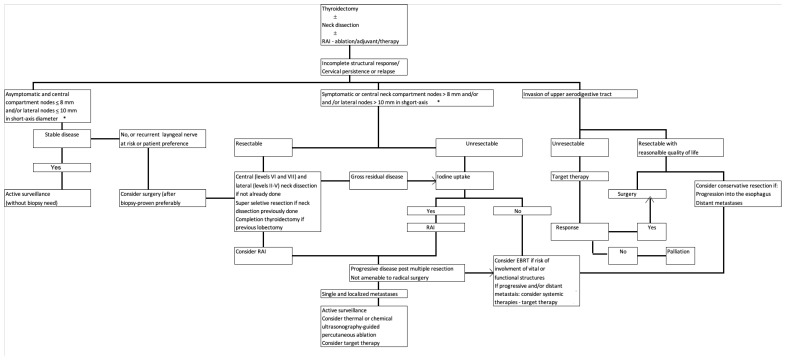
Recommendations for the management of the recurrent neck in well-differentiated thyroid carcinoma. * According to the American Thyroid Association guidelines [1]. RAI: Radioactive iodine. EBRT: External beam radiotherapy. Repeating RAI administrations after a cumulative activity of 600 mCi should be considered on per-patient basis [3]. High-risk patients or in the presence of active disease should receive suppressive doses of levothyroxine to maintain serum TSH levels below 0.1 mIU/mL, except in patients with clinical contraindications.

**Table 1 cancers-15-00923-t001:** Variables under initial concern to guide the basis of therapeutic decision making for patients with locoregional persistent or recurrent WDTC following initial treatment.

Variables Related to the Patient	Age	Variables Related to the Tumor	Initial Staging (AJCC-UICC TNM) [10]
	Comorbidities		Tumor histological subtype
	Life expectancy		Possibility of undifferentiated carcinoma
	Previous treatments related or not with thyroid tumor		Disease extent
	Symptoms		Size and location of the lesions
	Initial risk classification (ATA IRSS) [1]		Rate of tumor progression
	Dynamic risk stratification [12]		Presence of molecular markers associated with aggressive behavior
	Worries, wishes, values		Iodine avidity
			^18^FDG avidity
Variables related to the treatment	Appropriateness of the initial surgical resection		Presence of druggable mutations for targeted therapy
	Permanent complication		Initial staging (AJCC-UICC TNM) [10]
	Response to initial treatment [1]		Tumor histological subtype
	Current surgical resectability		Possibility of undifferentiated carcinoma
	Expected sequelae of the current therapy		

Legend: ATA IRSS: The American Thyroid Association Initial Risk Stratification Risk [1]; ^18^FDG: [^18^F]2-fluoro-2-deoxy-D-glucose.

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
