# Peer review of "Management of Recurrent Well-Differentiated Thyroid Carcinoma in the Neck: A Comprehensive Review†"

_cancers, 2023, doi:10.3390/cancers15030923_

Round 1

Reviewer 1 Report

Comment to the Authors

The manuscript in question represents a review focused on the management of loco-regional recurrence in well-differentiated thyroid cancer (WDTC). The Authors have been able to produce a comprehensive and well-written review that could be very helpful for clinicians.

In my opinion, however, some issues should be acknowledged before publishing it.

Major issues:

-          Page 4, “Recommended diagnostic work-up” section; in my opinion the sentence “If there is suggestion of progressive disease, total body radioactive iodine scanning is indicated to help therapeutic decisions based on RAI avidity and possible treatment of that disease if no further surgery is recommended” should be modified. The 2022 ETA Consensus Statement for the indications for post-operative RAI ablation [PMID: 34981741] (Recommendation 7) underlines that RAI WBS has a low impact on therapeutic decision-making and may be associated with stunning of thyroid remnant tissue and metastases; I would advise the Authors to include this statement in the paragraph and, therefore, modify the sentence accordingly.

-          Page 11, “Ultrasonography-guided percutaneous ablation” section; according to Recommendation 15 of 2019 ETA guidelines for advanced radioiodine-refractory DTC, I would add that these techniques could also be employed during systemic therapy, usually with an adequate and temporarily MKI withdrawal

-          Page 12-13, “Systemic therapy” section; the Authors should acknowledge the following issues:

o   I would add to the paragraph a brief overview of other FDA-approved drugs, like cabozantinib and RET inhibitors (selpercatinib/pralsetinib).

o   I would underline that there is no evidence (at least at the moment) regarding the use of systemic therapy in patients with aerodigestive invasion from locoregional recurrence in the absence of distant metastatic disease

o   I would strenghten the concept of multidisciplinary discussion in the setting of systemic therapy management

o   Please underline the fact that systemic therapy, as mentioned in the 2019 ETA guidelines, could be associated with other treatments and procedures (surgery, EBRT, ablation) in order to optimize disease control

-          I would advise the Authors to conduct a thorough revision of the correct order of bibliographic references, since, in some cases, the reference number does not correspond to the right citation (e.g. page 10, “Radioactive iodine” section; sixth to last line: the reference [14] should probably refer to the article of Fugazzola et al. that is however indexed as n°13)

Minor issues:

-          Table 1; I would add to the “Variables related to the tumour” section, the following line: - Presence of druggable mutations for targeted therapy

-          Page 9, “Radioactive iodine” section, first line of the paragraph; I would add “Adjuvant”, since the Authors are referring to post-surgical RAI treatment according to ATA risk of recurrence

Author Response

Reviewer 1

We appreciate your careful and valuable comments and we have tried to adapt the text to your suggestions.

The English language and style were revised. However, since we are authors from different nationalities, even if some are native English speakers, we cannot always agree on the same language style.

The note about RAI scanning according to the 2022 ETA Consensus Statement (new reference number 15) was added, and this reference has been cited on other relevant occasions. The same was done for the remark about page 11.

“…there is no evidence regarding the use of systemic therapy in patients with aerodigestive invasion from locoregional recurrence in the absence of distant metastatic disease” has been added as per your suggestion.

The participation of a multidisciplinary team in the context of systemic therapy is quite relevant. Thank you for the comment.

A note about the possibility of associating systemic therapy with other local therapeutic modalities was already on page 14, but the idea was reinforced according to your suggestion and the reference cited.

Still regarding systemic therapy, we focused on the approved drugs for WDTC. We have thus added new references.

Thank you also to notice the references mistakes.

We hope we have provided a satisfactory review, and again, thank you for your consideration.

Reviewer 2 Report

This is a well-written, information-rich and interesting review on different treatment modalities for recurrent well-differentiated thyroid cancer. It summarizes the recommendations from different thyroid associations as well as fairly recent studies published on the topic. I have a few comments that might improve the text.

1. Simple Summary is too similar to the Abstract, almost the same sentences are being used. I would rewrite either one of these sections, so that Simple Summary provides a rough outline of the review, and that Abstract gives new information on the topics of the review.

2. The text should be structured so that the reader knows which section is independent and which one is a subsection. Therefore, I suggest including numbers for sections and subsections, or different formatting between the headings for sections and subsections. Also, I feel that sections that start with the heading Surgery should be grouped into one section named for example Current treatment modalities and 6 sections from Surgery to Systemic Therapy should be subsections under that heading. The latter is just a suggestion, the authors should group the review section according to their wish, however I feel that some kind of numbered/formatted structure is necessary.

3. The (sub)section Surgery could be somewhat shortened, it is too long at this point.

4. In the Conclusions I would add a line or two to summarize what other options other than surgery followed by RAI ablation exist and when they can be used, as a great part of the article is about other treatments.

5. Most of the references date from before the year 2020. Are there more recent studies regarding the management of WDTC, particularly considering treatment modalities other than systemic therapy?

Author Response

Reviewer 2

We appreciate your careful and valuable comments and we have tried to adapt the text to your suggestions.

The Simple Summary was modified as suggested.

We admit that the Surgery section is long, but we justify it as the surgical therapy is the treatment of choice and associated with several variables. The authors are also surgeons, and this is the field in which we have the greatest intimacy and contribution. Likewise, surgical therapy was privileged in the Conclusion section.

We thank you for pointing out this “bias” and have tried to tailor the sections.

The therapeutic modalities have been organized in subsections under Current treatment modalities.

Recent references were introduced, however, an initial thorough selection of more than 100 articles was made and the date of publication was only one of the criteria used. Those less recent were kept because of their historical as well as practical importance in the management of these cases.

Again, thank you for your valuable contribution.

Reviewer 3 Report

  • The Idea in general is unique and good 
  • Adding more diagrams and pic to the paper will make it more attractive 
  • I think it more better to be a book chapter ( Keep in consideration)
  • Need more new references only 6-7 were >2020

Author Response

Reviewer 3

We appreciate your careful and valuable comments and we have tried to adapt the text to your suggestions.

We admit that the article is long, as the topic in question also admits several considerations. We thank you for the idea of a book chapter.

Recent references were introduced, however, an initial thorough selection of more than 100 articles was made and the date of publication was only one of the criteria used. Those less recent were kept because of their history as well as practical importance in the management of these cases.

Again, thank you for your valuable contribution.

Reviewer 4 Report

The authors describe the correct analyse and relevant dissscusion. 
only one question can be associte : are the deescalation surgery rules today relevant or will need some correction? 
on the second hand the concept of personal tailoring therapy is relevant to risk factors and genes analyse and can be an use in clinical practice today?? 

Author Response

Reviewer 4

We appreciate your careful analysis.

Two very relevant issues were pointed out and shall be addressed in the future. In fact, the operation has not been de-escalated for cervical recurrences, but there is an opportunity to no longer indicate surgical treatment when its benefit is uncertain. However, when opting for surgery, the goal is to obtain neoplasm-free margins.

Personal tailoring therapy should be our primary goal, hence the importance of multidisciplinary discussions between experienced peers. Genetic analysis, however, is not routinely performed to date.  However, for some selective systemic therapies, such as selpercatinib, the presence of the mutation in question, in this case, RET, is ideal. In the future, in turn, we hope that genetic data will guide us from the first therapy.

Again, thank you for your valuable contribution.

Reviewer 5 Report

Thank the Editor to give me the opportunity to revise this article.

I read this article with great interest. This is a comprehensive review of the literature, carried out with the aim of exploring and deepening the current knowledge on the management of differentiated thyroid cancer local recurrence.
The manuscript is very well written. The work is well structured, clear in all its parts. The topics covered are consistent with each other. The conclusions are well argued. The literature underlying the review is large and complete. The schemes are clear and well structured and provide a relevant contribution to the whole article.

In conclusion, the work is of great impact and excellent quality and is ready for publication.

Author Response

Reviewer 5

We appreciate your comments and compliments on the article. We really hope to contribute with this review. Again, thank you for your words and encouragement
